# Interpretable Emotion Attribution in Social Graphs: A Comparative Analysis of Rule-Based, Transformer, and LLM Models

**Usman Gidado & Anton Kolonin**
Department of Mathematics & Mechanics
Novosibirsk State University
Novosibirsk, Russia
{ubgidadoac,akolonin}@gmail.com

## Abstract

Emotion attribution in social graphs requires inferring directed emotional attitudes between entities in complex, multi-turn dialogues. While transformer models dominate the field, they often lack the transparency required for social science applications. We present a systematic comparison of three modeling paradigms for this task: a fully interpretable rule-based system, a fine-tuned RoBERTa-large model, and a few-shot Llama-3-8B. Utilizing the DialogRE dataset, we demonstrate that incorporating a 3-turn conversational context significantly improves attribution accuracy across paradigms. Crucially, our results show that the interpretable rule-based system achieves a competitive F1 score, making it statistically indistinguishable from the state-of-the-art RoBERTa model. In contrast, we find that few-shot large language models exhibit poor performance in emotion attribution of semantic relations, performing below the rule-based baseline. We prove that interpretability does not necessitate a performance trade-off in social emotion analysis.[1] [2]

## 1 Introduction

Emotion Recognition in Conversation (ERC) has primarily focused on identifying the emotional state expressed by an individual speaker at a given time (Khalid & Sano, 2023). While this formulation is effective for monologic or dyadic sentiment analysis, it is insufficient for complex multi-party interactions, where emotions are often directed toward specific entities rather than expressed in isolation. In such settings, understanding who feels what toward whom becomes essential. This problem, commonly referred to as emotion attribution, naturally induces a social graph representation, where the entities form nodes and the directed edges encode emotional relations between them (Dekker et al., 2019; Khemani et al., 2024).

Emotion attribution over social graphs is increasingly relevant in real-world applications which include social media analysis, organizational communication, mental health monitoring, and legal or forensic dialogue interpretation. In these domains, predictive accuracy alone is not sufficient: systems must also support interpretability, allowing practitioners to inspect, validate, and correct model decisions when necessary (Tan et al., 2025; Sarma, 2019). This requirement is particularly acute in regulated or high-stakes environments, where opaque decision-making can hinder trust, accountability, and deployment. Despite these constraints, the dominant approaches to emotion attribution in recent NLP research rely on black-box neural architectures, most notably fine-tuned Transformer encoders and, more recently, large language models (LLMs) used in few-shot or zero-shot settings (Lalk et al., 2025; Mohammadi et al., 2025). While these models often achieve strong empirical performance, they provide limited insight into why a particular emotional relation is predicted, and inability to trace its decision back to a specific linguistic trigger, this prevents practitioners from performing systematic error correction. In contrast, rule-based approaches, which offer full

---

[1]**Code:** https://github.com/UBGidado/Interpretable-EA-in-SG
[2]**Model:** https://github.com/aigents/pygents/tree/main/pygents

transparency and explicit reasoning, are often dismissed as outdated or fundamentally less capable, particularly in comparison to large-scale transformer models.

In this work, we revisit this assumption through a comparative study of emotion attribution in social graphs, explicitly examining the trade-offs between performance and interpretability across rule-based, Transformer, and few-shot LLM approaches under a unified relational representation. While LLMs demonstrate remarkable zero-shot capabilities, it remains unclear whether few-shot prompting can match the calibrated decision boundaries of smaller, task-specific fine-tuned models in nuanced domains. This question is particularly relevant given the growing reliance on in-context learning. However, systematic experiments with Llama-3-8B-Instruct yield significantly lower performance, achieving a maximum Macro-F1 of 0.528 (10 few-shot) and 0.501 in our prompt ablation study. A systematic ablation across multiple prompt templates and shot counts produces F1 scores ranging from 0.294 to 0.501, indicating that prompt engineering alone cannot close the gap with fine-tuned models. In contrast, our interpretable rule-based system achieves performance statistically comparable to fine-tuned Transformers, challenging the assumption that interpretability necessarily requires sacrificing accuracy. However, our evaluation is limited to DialogRE; generalization to other dialogue domains requires further study.

Our work makes three key contributions to interpretable emotion attribution in social graph:

1. We demonstrate that interpretable rule-based approaches achieve statistically competitive performance with fine-tuned transformers (see Table 2), challenging the assumption that explainability necessarily sacrifices substantial accuracy. This finding has important implications for deploying emotion attribution systems in regulated domains requiring transparency.

2. We show that few-shot prompting of large language models (Llama-3-8B) significantly underperforms targeted fine-tuning (see Table 2), even with sophisticated prompt engineering. This counter-intuitive result demonstrates that task-specific adaptation remains essential despite recent advances in foundation model scale and capabilities.

3. We provide the systematic comparison of explainability–performance trade-offs across three paradigms; rule-based, neural, and LLM approaches with statistical validation. This enables informed decisions about when interpretability can be prioritized without sacrificing performance.

## 2 RELATED WORK

### 2.1 INTERPRETABILITY IN NLP

Interpretability in NLP is divided into post-hoc explanation and inherent transparency. Post-hoc methods, such as LIME and SHAP (Ribeiro et al., 2016; Lundberg & Lee, 2017), approximate the behavior of black-box models by estimating feature importance. Despite their widespread adoption, these approaches are often criticized for their dependence on the underlying machine learning model and their vulnerability to feature collinearity, underscoring the need for caution in their application and interpretation to prevent misleading analyses. (Salih et al., 2024). In contrast, inherently interpretable models are transparent by design. This category includes rule-based systems, decision trees, and weighted n-gram models. The Aigents framework exemplifies this paradigm by employing weighted structured patterns and heterarchical n-grams for multi-factor classification (Kolonin & Arinicheva, 2025), enabling explicit inspection and modification of decision logic. Such transparency is particularly valuable in domain-specific or high-stakes settings, where bias correction and rule-level validation are required (Salih et al., 2024).

More recently, mechanistic interpretability has emerged as an effort to reverse-engineer neural models by identifying neurons or circuits associated with linguistic concepts (Rai et al., 2024). These methods are promising, but remain computationally intensive and difficult to audit without specialized expertise (Sharkey et al., 2025). This tension between the generalization strength of neural models and the auditability of symbolic systems has driven the development of hybrid approaches that integrate neural pattern recognition with rule-based validation (Kolli, 2025).

Interpretable NLP models, such as rule-based or symbolic systems, have long been valued for transparency, especially in high-stakes tasks (Kolonin & Arinicheva, 2025). For example, recent work

by Lalk et al. (2025) applies step-by-step reasoning in sentiment analysis. However, such explanations often trade factual accuracy for interpretability. Our rule-based Aigents system follows this paradigm. Unlike prior ERC work that focuses on improving accuracy with complex neural architectures (Kim & Vossen, 2021), we explicitly examine how a fully interpretable system compares to opaque models, quantifying the interpretability–performance trade-off in relational emotion attribution. This allows for a nuanced understanding of model behavior, crucial for domains where accountability and trustworthiness are paramount (Tak et al., 2025). For instance, directed acyclic graph networks and graph neural networks have demonstrated improved performance in modeling conversational context for emotion recognition, yet their interpretability often remains limited (Khemani et al., 2024; Shen et al., 2021).

## 2.2 LLM Data Augmentation

Large Language Models have recently been leveraged to generate or label additional data in low resource settings like ours (Zhang et al., 2023b). GPT-based models can synthesize training examples by prompt-based few-shot generation (Kaddour & Liu, 2023), and have been used to create dialogue or summary data for related tasks. Feng et al. (2021) use DialoGPT to annotate summarization features in conversational text. And more generally, Zhang et al. (2023a) introduce an active learning framework (LLMaAA) in which an LLM acts as an annotator to label unlabeled inputs, improving classification with minimal human cost. Such LLM-augmented data has been shown to boost performance on complex tasks. In our work, we use few-shot LLaMA-3 prompts to generate emotion-label examples from our training set, following this line of research. We compare model performance with and without LLM-augmented examples, studying how data augmentation interacts with interpretability.

## 2.3 Social Graph Emotion Attribution

Recent work has begun to incorporate relational and graph structure into emotion prediction. Yu et al. (2020) introduced the DialogRE dataset for dialogue-based relation extraction, emphasizing speaker-aware modeling to capture cross-utterance relations . Graph-based ERC models extend this idea: for example, Liu et al. (2023) build heterogeneous conversation graphs to propagate contextual cues. These approaches improve emotion recognition in multi-party dialogues by explicitly modeling interactions. Similarly, in social media contexts, researchers have applied GNNs to predict users' emotions from text and network structure.For instance, an enhanced Text-GCN model employing a Pointwise Mutual Information (PMI)–based graph construction strategy achieves high emotion classification performance (78.64% and 92.38% accuracy) on Twitter datasets, while incorporating interpretability components such as attention-based word importance (Khemani et al., 2025). These explanations remain primarily post-hoc and model-internal. They do not provide auditable reasoning over relational emotional attributions between entities. In contrast, our work focuses on inherent interpretability at the level of social relations, enabling direct inspection and modification of emotion attribution logic across graph edges. Furthermore, our study is novel in comparing interpretable (rule-based) and black-box (transformer or LLM) approaches on this graph-based emotion attribution task. This allows us to assess how model design affects both performance and explainability in a relational emotion setting.

## 3    Problem Formulation

We formulate emotion attribution as a relation extraction task over multi-turn dialogues. Given a dialogue and an entity pair (subject, object), the goal is to classify the emotional relationship from subject to object as positive, negative, or neutral based on conversational context.

## 3.1 Task Definition

Following the dialogue-based relation extraction framework of Yu et al. (2020), we define our task formally as follows. Given a dialogue $D = \{(s_1, t_1), (s_2, t_2), \ldots, (s_m, t_m)\}$ where $s_i$ denotes the speaker identifier and $t_i$ represents the utterance text of the $i$-th turn, and an entity pair $(e_{\text{subj}}, e_{\text{obj}})$, the task is to predict the relation type $r \in \mathcal{R}$ from $e_{\text{subj}}$ to $e_{\text{obj}}$ based on $D$, where $\mathcal{R} = \{\text{positive}, \text{negative}, \text{neutral}\}$.

Formally, the objective is to learn a function

$$f : (D, e_{\text{subj}}, e_{\text{obj}}) \rightarrow \mathcal{R},$$

which maps a dialogue and an ordered entity pair to an emotion label.

**Entity Representation:** We represent entity pairs using special marker tokens. For an entity pair $(e_{\text{subj}}, e_{\text{obj}})$, we surround each entity mention in the dialogue with special tokens: `[SS]` $e_{\text{subj}}$ `[/SS]` for the subject and `[OS]` $e_{\text{obj}}$ `[/OS]` for the object. This enables models to attend to relevant entity mentions throughout the conversation. For speaker entities, we mark the speaker identifier at the beginning of each turn. The concatenated dialogue with entity markers serves as input to relation extraction models.

## 3.3 DATASET

We utilize DialogRE v2 (Yu et al., 2020), a first human-annotated dialogue-based relation extraction dataset derived from *Friends* TV show transcripts, consisting of 1,073 training, 358 development, and 357 test dialogues. The dataset contains 10,168 relation instances spanning 36 relation types, which we organize into five high-level categories: Identity & Metadata (45.0%), including alternate names and unanswerable cases; Personal Relationships (16.1%), such as friends (7.1%) and siblings (3.1%); Interpersonal Sentiment via positive (7.6%) and negative (2.6%) impressions; Location & Geography (4.4%); and Professional & Education (4.3%) relations. DialogRE exhibits several properties that make it particularly challenging and suitable for studying explainability–performance trade-offs: 96.0% of relations require cross-sentence reasoning, 89.9% involve at least one speaker entity, and the conversational style is characterized by high pronoun usage and low information density, which also complicates relation extraction in multi-turn contexts. These characteristics make DialogRE an effective benchmark for evaluating whether interpretable models can maintain competitive performance on complex, real-world conversational data.[3]

**Relation Extraction Model:** To obtain entity pairs and their contextual windows for emotion classification, we employ a DeBERTa-v3-large model fine-tuned on the DialogRE dataset. DeBERTa-v3-large has demonstrated strong performance on dialogue understanding tasks due to its disentangle attention mechanism and enhanced mask decoder (He et al., 2021). We fine-tune the model to identify subject–object entity pairs and extract their corresponding 3-turn conversational context (preceding turn, target turn, and following turn). These extracted tuples serve as structured inputs to all downstream emotion attribution models, enabling a controlled comparison across modeling paradigms.

**Training Data:** For neural model training, we adopt an LLM-assisted silver-labeling strategy. We employ Llama-3-8B-Instruct to annotate 1,179 instances from the DialogRE training split. The model is prompted with the full dialogue, the target entity pair, and explicit emotion-relation definitions.To guide the model's predictions, we include 150 manually annotated instances as in-context demonstrations.To ensure label quality, we retain only predictions with confidence scores $\geq 0.70$. This process yields a balanced silver-labeled training set comprising 400 positive, 393 neutral, and 386 negative samples. We further reserve 20% of this data for validation during fine-tuning.

**Test Set Construction:** From the official DialogRE test split, we extracted 150 entity pairs using to ensure a controlled evaluation environment. To maintain focus on interpersonal dynamics, we filtered the data to include only recognized social relations, such as *per:friends*, *per:spouse*, and *per:girl/boyfriend*, while excluding non-social attributes like *per:age* or *per:title*. For each sampled pair, we extracted a 3-turn contextual dialogue window (comprising the target utterance and its immediate neighbors) to provide the situational evidence required for manual annotation and rule-based sentiment analysis. The final gold-standard distribution for this test set consists of 70 positive (46.7%), 44 negative (29.3%), and 36 neutral/unanswerable (24.0%) instances, providing a benchmark for comparing neural fine-tuning against our interpretable Aigents approach.

---

[3]`https://github.com/nlpdata/dialogre`

# 4 MODELS

We evaluate three distinct paradigms for emotion attribution, each representing a different point along the explainability-performance spectrum: rule-based pattern matching (Aigents), fine-tuned neural models (RoBERTa), and few-shot large language models (Llama-3).

## 4.1 RULE-BASED APPROACH: AIGENTS

Aigents is an interpretable n-gram matching system originally developed for sentiment analysis (Raheman et al., 2022). The model operates through dictionary-based pattern matching with predefined lexicons of emotional expressions. We adapt Aigents for dialogue-based emotion attribution by implementing cross-turn pattern aggregation with exponential strength normalization.

**Architecture:** Aigents employs a hierarchical n-gram matching strategy with the "priority on order" principle: longer n-grams take precedence over their constituent shorter n-grams. For instance, if the tetragram ["not", "a", "bad", "thing"] matches, all constituent n-grams including the bigram ["bad", "thing"] and unigram ["bad"] are disregarded. This prevents double-counting and allows the model to capture negation and multi-word expressions accurately. Similarly, matching the bigram ["no", "good"] disregards both constituent unigrams ["no"] and ["good"]. The system maintains separate lexicons for positive and negative emotional expressions. The positive lexicon comprises over 3,800 n-grams, and the negative lexicon contains over 8,200 n-grams,

**Scoring Algorithm:** For each dialogue turn, Aigents extracts all matching n-grams and computes separate positive ($P$) and negative ($N$) raw scores based on match frequencies. The model then applies an exponential strength transformation to normalize scores:

$$r = \max(P, |N|) \tag{1}$$
$$s = 1 - e^{-1.5 \cdot r} \tag{2}$$

where $r$ represents the raw dominant score and $s$ is the normalized strength in range [0, 1]. This exponential transformation maps raw scores as follows: $0.3 \rightarrow 0.36$, $0.5 \rightarrow 0.53$, $1.0 \rightarrow 0.78$, and $1.5 \rightarrow 0.89$. For texts containing multiple emotion words (total emotion $> 1.0$), we apply an optional multi-word boost:

$s' = \min(1.0, s + \min(0.15, (P + |N| - 1.0) \times 0.1))$.

**Classification Decision:** The model computes a margin score $m = |P - |N||$ and confidence $c = \frac{m}{P+|N|}$ when the denominator is non-zero. The final three-way classification uses threshold-based rules:

- **neutral** if $m < \tau_m$ and $r < \tau_r$
- **positive** if $P > |N|$ (among non-neutral cases)
- **negative** if $|N| > P$ (among non-neutral cases)

where $\tau_m = 0.2$ (margin threshold) and $\tau_r = 0.3$ (raw strength threshold) are empirically tuned on validation data. This confidence-based approach enables the inherently binary Aigents model to detect neutral cases when both positive and negative signals are weak or balanced.

**Neutral Label Handling for Aigents Evaluation**: Aigents performs binary classification (positive, negative), while gold labels include a neutral class. To enable fair comparison, we use three evaluation modes. *Mode 1* excludes gold neutral instances, evaluating binary performance on polarized samples only (N = 114). *Mode 2* maps neutral labels to the dorminant class class, enabling full dataset comparison (N = 150). *Mode 3* excludes samples identified as likely neutral based on weak lexical evidence, focusing on high-confidence predictions (N = 97). These complementary modes ensure fair evaluation while reflecting the architectural constraints of rule-based emotion attribution.

**Contextual Integration:** To capture the emotional nuances of multi-turn conversations, we analyze a 3-turn context window (preceding, target, and following utterances) as a unified situational block. Rather than applying arbitrary linear weights to each turn, we aggregate all n-gram matches across

the window into a total raw score $R$. This holistic approach ensures that supporting evidence (such as a prompt in a preceding turn or a reaction in a following turn) is captured with equal fidelity.

The aggregate signal is then transformed via an exponential saturation function:

$$S(R) = 1 - e^{-1.5R} \tag{3}$$

to determine the final relationship strength $S(R)$. This non-linear transformation prevents "evidence inflation" while maintaining the interpretability of the extracted emotional signals.

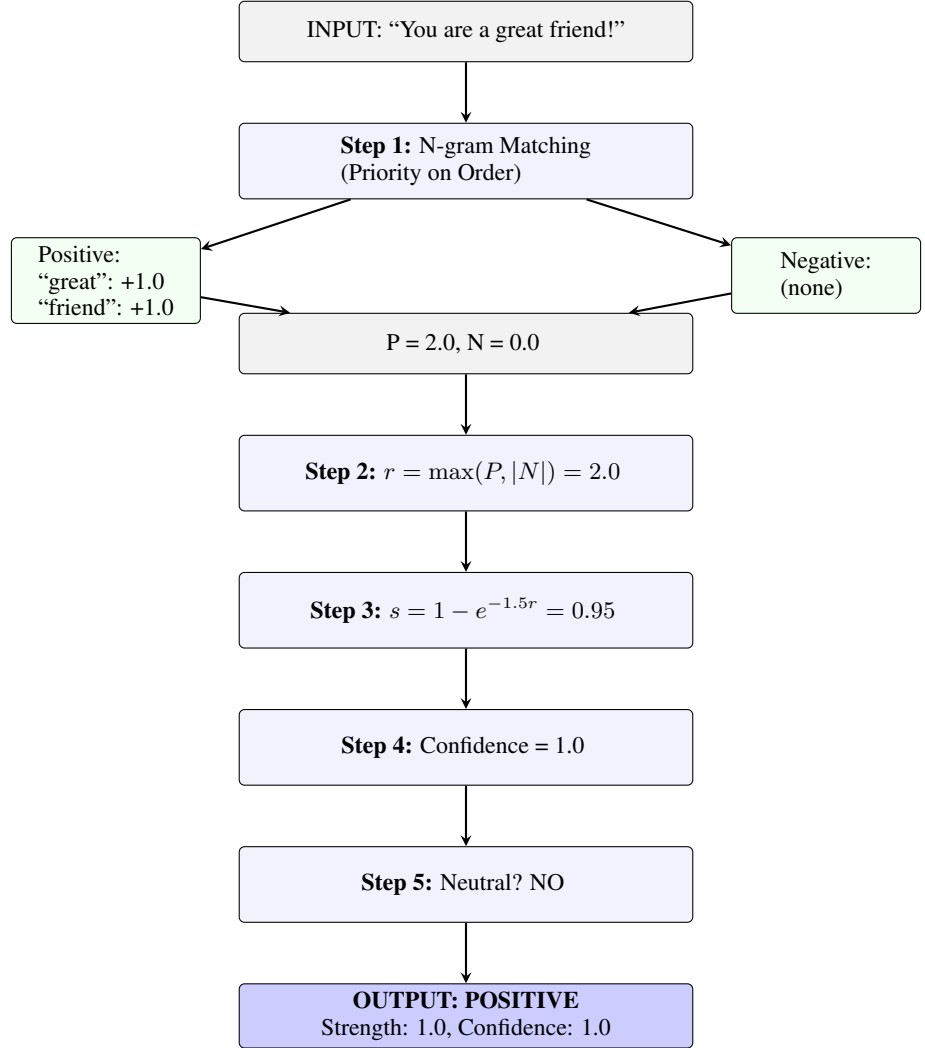

Figure 1: Interpretable emotion attribution pipeline of Aigents. The model applies ordered n-gram matching over positive and negative lexicons, followed by exponential strength normalization and confidence-based neutral detection, yielding a fully transparent polarity decision.

### 4.2 FINE-TUNED NEURAL APPROACH: ROBERTA:

We employ ROBERTA (Liu et al., 2019), a robustly optimized Transformer encoder pre-trained on 160GB of text with dynamic masking and extended training. To incorporate emotion-specific prior knowledge, We initialize from a RoBERTa-large checkpoint fine-tuned on GoEmotions (Demszky et al., 2020), publicly available via the HuggingFace Model Hub.[4], a large-scale emotion annotation

---

[4]https://huggingface.co/Laksssssshya/RoBERTa-large-goemotions

dataset comprising approximately 58k Reddit comments labeled across 28 fine-grained emotions and a neutral class. The GoEmotions pretrained checkpoint we adopt was optimized for multi-label emotion classification using focal loss and label-specific threshold calibration to address class imbalance. While the original model supports multi-label prediction, we repurpose the encoder representations for single-label emotion attribution by replacing the original output layer with a task-specific classifier trained on DIALOGRE-derived emotion labels.

**Context Variants:** To assess the impact of conversational context, we evaluate two input configurations: ROBERTA-1TURN (`[CLS] target_turn [SEP]`) processes only the target utterance containing the subject–object relation, while ROBERTA-3TURN (`[CLS] turn_1 [SEP] turn_0 [SEP] turn_{+1} [SEP]`) incorporates a local conversational window consisting of the preceding, target, and following turns. Both variants use a maximum sequence length of 257 tokens, covering approximately 95% of DIALOGRE instances without truncation.

### 4.3 FEW-SHOT LLM APPROACH: LLAMA-3

We evaluate Llama-3-8B-Instruct (AI@Meta, 2024), an instruction-tuned large language model with 8 billion parameters trained on over 15 trillion tokens. We evaluate it using in-context few-shot learning with a baseline prompt consisting of detailed task instructions and 10 labeled examples. To determine whether performance limitations arise from sub-optimal prompting rather than inherent model constraints, we conduct a systematic ablation across three prompt templates and three shot counts (3, 5, and 10 examples), yielding 9 configurations. The evaluated prompt designs include: (1) a standard instruction-based template with explicit classification guidelines, (2) a chain-of-thought template encouraging step-by-step reasoning, and (3) a structured JSON-format template enforcing constrained outputs (Table 3).

## 5 EXPERIMENTAL SETUP

**Relation Extraction (DeBERTa-v3-Large):** For relation extraction on DialogRE, we fine-tune microsoft/deberta-v3-large model [5] with a maximum sequence length of 512. Due to memory constraints, we use a per-device batch size of 2 with gradient accumulation over 4 steps (effective batch size of 8). The model is trained for 5 epochs using a learning rate of $1 \times 10^{-5}$. To address class imbalance, we downsample 70% of *no_relation* instances and apply a reduced loss weight (0.5) to the *unanswerable* class. All experiments are conducted with fixed random seeds (42) to ensure reproducibility.

**Neural Emotion Attribution (RoBERTa):** As discussed in section 4, we fine-tune two variants of RoBERTa-large, initialized from the GoEmotions-specialized checkpoint.We employ the AdamW optimizer with a learning rate of $8 \times 10^{-6}$ and $9 \times 10^{-6}$ for 1turn and 3turns model respectively, weight decay of 0.01, and a batch size of 16. To prevent overfitting on the dialogue context, we implement early stopping with a patience of 3 epochs based on the validation Macro F1 score. Training is conducted on an NVIDIA T4 GPU using mixed-precision (FP16), typically reaching convergence within 3–5 epochs.

**LLM-Based Data Augmentation (Llama-3):** To generate silver labels, we employ Llama-3-8B-Instruct[6] with 4-bit NormalFloat4 (NF4) quantization and double quantization. This reduces the memory footprint from ~15GB to ~5.3GB. Inference is performed via greedy decoding (temperature = 0.1, max 256 tokens) with a 10-sample few-shot strategy.

**Evaluation Metrics:** We evaluate performance using the Macro-averaged F1 score to balance representation across positive, negative, and neutral classes. We report results for RoBERTa (1-turn), RoBERTa (3-turn), and Aigents. Statistical significance is verified via 95% confidence intervals derived from bootstrap resampling ($N = 1000$).

---

[5]`https://huggingface.co/microsoft/deberta-v3-large`
[6]`https://huggingface.co/meta-llama/Meta-Llama-3-8B-Instruct`

Table 1: Hyperparameter specifications across model architectures.

| Hyperparameter | RoBERTa (1T) | RoBERTa (3T) | Llama-3 | Aigents |
|---|---|---|---|---|
| *Model:* | | | | |
| Base | RoBERTa-large | RoBERTa-large | Llama-3-8B-Instruct | N-gram Lexicon |
| Init | GoEmotions FT | GoEmotions FT | Meta | Pre-built |
| Quant. | None | None | 4-bit NF4 | None |
| Max Len | 128 | 256 | 2048 | N/A |
| *Training:* | | | | |
| Optimizer | AdamW | AdamW | N/A | Deterministic |
| LR | $8 \times 10^{-6}$ | $9 \times 10^{-6}$ | N/A | N/A |
| WD | 0.01 | 0.01 | N/A | N/A |
| Batch | 16 | 16 | 1 | N/A |
| Epochs | 15 | 15 | N/A | N/A |
| *Logic:* | | | | |
| Temp | N/A | N/A | 0.1 | N/A |
| Formula | Linear | Linear | N/A | $1 - e^{-1.5R}$ |
| Thresh | Softmax | Softmax | N/A | 0.4 |
| *Env:* | | | | |
| Hardware | T4 (Colab) | T4 (Colab) | T4 (Colab) | CPU (Docker) |
| Precision | FP16 | FP16 | 4-bit | FP32 |

Table 2: Overall performance comparison of rule-based (Aigents), fine-tuned transformer (RoBERTa), and few-shot LLM (Llama-3) approaches for emotion attribution in social graphs. RoBERTa-3turn achieves the highest Macro F1 (0.636), demonstrating the benefit of multi-turn conversational context. The interpretable Aigents system achieves comparable performance (Macro F1 up to 0.58, 0.6236, 0.625 when neutral are mapped to dorminant class, when gold neutral are excluded, and when detected neutral are excluded respectively). Few-shot Llama-3 exhibits substantially lower and less stable performance (Macro F1 0.528). Confidence intervals (95%) are computed via bootstrap resampling ($N = 1000$).

| Model | N | Acc | F1 Pos | F1 Neg | F1 Neu | F1 Macro [CI] |
|---|---|---|---|---|---|---|
| RoBERTa-3turn | 150 | 0.653 | 0.696 | **0.719** | 0.493 | **0.636 [0.554, 0.715]** |
| Aigents (Mode 3)†† | 97 | **0.701** | **0.794** | 0.453 | — | 0.624 [0.516, 0.727] |
| Aigents (Mode 2)† | 150 | 0.593 | 0.651 | 0.512 | *mapped* | 0.582 [0.500, 0.659] |
| Aigents (Mode 1) | 114 | 0.667 | 0.750 | 0.500 | — | 0.625 [0.526, 0.712] |
| Llama-3§ | 150 | 0.567 | 0.642 | 0.537 | 0.407 | 0.528 [0.484, 0.568] |
| RoBERTa-1turn | 150 | 0.633 | 0.676 | 0.638 | **0.537** | 0.617[0.579, 0.655] |
| RoBERTa-baseline | 150 | 0.573 | 0.708 | 0.500 | 0.405 | 0.538 [0.496, 0.580] |

## 6 RESULTS

### 6.1 OVERALL PERFORMANCE

Llama-3-8B achieves F1 macro = 0.528, substantially lower than RoBERTa-3turn (0.636), indicating that few-shot prompting without task-specific fine-tuning provides insufficient performance gain. Ablation across 9 configurations (Table 3, Appendix) shows performance ranges from F1 = 0.501 (best: Standard template, 3-shot) to F1 = 0.294 (Chain-of-Thought, 5-shot), indicating that prompt engineering provides marginal improvements but cannot overcome inherent LLM limitations on this task.

### 6.2 STATISTICAL SIGNIFICANCE

We assess statistical reliability using 95% bootstrap confidence intervals computed from 1000 re-samples (Table 2). RoBERTa-3turn achieves the highest macro F1 (0.636, CI: [0.554, 0.715]), slightly outperforming RoBERTa-1turn (CI: [0.579, 0.655]). The small overlap between inter-

vals suggests a modest but consistent benefit from incorporating multi-turn conversational context. In contrast, RoBERTa-3turn substantially outperforms the few-shot Llama-3 model (CI: [0.484, 0.568]), with non-overlapping confidence intervals indicating a statistically significant difference ($p < 0.05$). Further analysis of nine prompting configurations for Llama-3 reveals high sensitivity to prompt design (F1 range: 0.294–0.501), yet even the best configuration remains well below RoBERTa-3turn and the interpretable Aigents Mode 3 system (F1 = 0.624). This gap, combined with performance degradation under alternative prompt strategies (e.g., Chain-of-Thought mean F1 = 0.328), highlights inherent limitations of few-shot prompting for structured emotion classification compared to task-specific fine-tuning and interpretable rule-based methods. Finally, all advanced approaches substantially outperform the baseline RoBERTa model, confirming the importance of multi-turn context and task-specific adaptation. These results challenge the conventional assumption that explainability necessarily entails performance trade-offs: Aigents achieves competitive performance with RoBERTa while providing complete transparency.

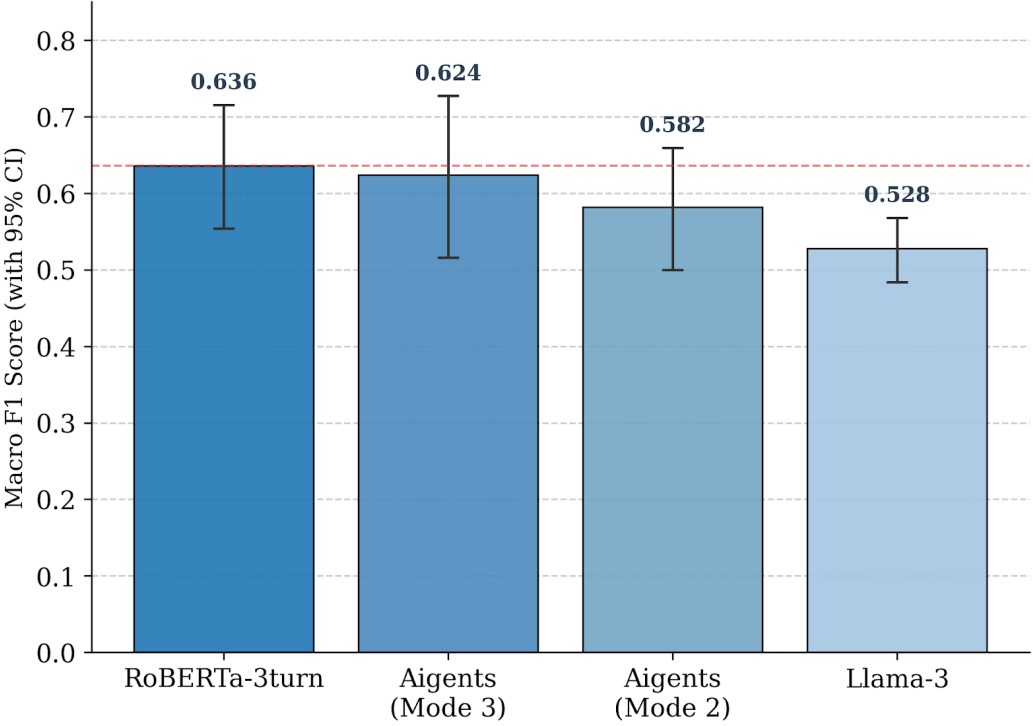

Figure 2: Overall model performance Macro F1 with 95% Bootstrap Confidence Intervals [CI]. Bold indicates the highest value in each metric. Aigents (mode 2) 3-class mode where Neutral instances were mapped to dorminant, Aigents (Mode 3) binary mode where detected neutrals excluded, and Llama-3 8B few-shot performance. The RoBERTa-3turn model achieves the highest Macro F1, while Aigents (Mode 3) demonstrates superior precision on its targeted subset.
.

## 7 DISCUSSION

Our results show that interpretability can be achieved without sacrificing performance. The interpretable Aigents system attains F1 = 0.582 (mapped) and F1 = 0.624 (detected neutral excluded) on binary emotion classification, statistically comparable to RoBERTa-3turn (F1 = 0.636) based on overlapping confidence intervals. Unlike neural models, Aigents provides explicit, human-readable

justifications through matched lexical patterns and deterministic weights, making its predictions directly auditable. However, Aigents does not support neutral emotion attribution, which constitutes 24% of the dataset, highlighting a key limitation of rule-based approaches in modeling absence-based categories. Consequently, interpretable systems are well-suited for transparent binary attribution, while fine-tuned neural models remain necessary for full multi-class coverage. Incorporating multi-turn conversational context yields a modest improvement over single-turn RoBERTa, with confidence intervals indicating a small but consistent benefit. This suggests that while most emotion cues reside within the target utterance, local conversational context provides additional disambiguating information in some cases. Given the increased computational cost of multi-turn inputs, this improvement represents a favorable trade-off in accuracy-critical applications.

Few-shot prompting with Llama-3 substantially underperforms both fine-tuned and rule-based approaches. Despite systematic evaluation across nine prompt configurations, Llama-3 achieves a maximum F1 of 0.528 (10 few-shot) and 0.501 with ablation, with confidence intervals significantly below RoBERTa-3turn and consistently lower than Aigents. The large performance variance across prompts (F1 range: 0.294–0.501) indicates sensitivity to prompt design, but even optimal prompting fails to close the gap. Notably, the interpretable Aigents system outperforms Llama-3 by approximately 0.12 F1, demonstrating that transparent, rule-based methods can exceed few-shot large language models on structured emotion attribution tasks where explicit lexical cues dominate. This suggests that the limitation lies in the few-shot paradigm itself rather than prompt engineering, and that task-specific fine-tuning is necessary for neural models to achieve competitive performance. Overall, these findings highlight three key conclusions: (1) interpretable rule-based systems can achieve performance comparable to fine-tuned neural models in binary settings, (2) multi-turn conversational context provides modest but consistent improvements, and (3) few-shot prompting alone is insufficient for high-accuracy emotion attribution without task-specific adaptation.

## 8 LIMITATIONS

Several limitations should be acknowledged. First, the evaluation is conducted on a relatively small test set (N = 150), which, while sufficient for bootstrap-based statistical comparison, limits the strength of generalization claims. Furthermore, the data are derived from scripted *Friends* dialogues, which may over-represent explicit emotional expressions and structured conversational patterns compared to spontaneous real-world interactions. Moreover, the Aigents system does not natively model neutral emotion, as its rule-based architecture detects emotions through the presence of explicit lexical cues. To enable fair three-class comparison, neutral instances were mapped using confidence-based procedures (Mode 2), which introduces an approximation and may not fully capture neutrality as an independent semantic category. Additionally, although silver labels were filtered using confidence thresholds and partial manual validation, residual labeling bias from the LLM-assisted annotation process may persist and influence model evaluation. Finally, few-shot Llama-3 evaluation, while systematically examined across nine prompt configurations, remains constrained by the inherent variability and instability of prompt-based inference. Although our ablation improves robustness of conclusions, it does not represent an upper bound achievable through full task-specific fine-tuning of large language models.

## 9 CONCLUSION

This work provides a controlled comparison of interpretable rule-based, fine-tuned neural, and few-shot LLM approaches for dialogue emotion attribution. On the shared 150-sample test set, RoBERTa-3turn achieves the highest overall performance (F1 = 0.636), while the interpretable Aigents system remains competitive (F1 = 0.582 in 3-class mode; F1 = 0.624 on high-confidence predictions), demonstrating that transparent rule-based models can approach neural performance when lexical emotion cues are explicit. In contrast, systematic ablation across nine prompting configurations shows that few-shot Llama-3 reaches a maximum F1 of 0.501, substantially below both Aigents and RoBERTa, indicating inherent limitations of few-shot prompting for structured emotion classification. These findings highlight a practical trade-off: rule-based systems offer full interpretability and competitive binary performance, while fine-tuned neural models provide superior multi-class coverage. Future work will explore hybrid architectures combining interpretable lexical reasoning with neural handling of neutral and ambiguous emotional expressions.

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

## A    APPENDIX

## B    TRAINING DYNAMICS ANALYSIS

The RoBERTa-3turn model exhibits standard fine-tuning behavior: rapid optimization during early training (steps 0–150), followed by convergence between steps 150–300. Validation loss attains its minimum at step 300, after which performance degrades, and early stopping is therefore applied to prevent overfitting. At this point, the model achieves a macro-F1 of 0.809, with particularly strong performance on positive emotion (F1 = 0.845). Neutral emotion remains the most challenging class (F1 = 0.779), suggesting that additional conversational context can introduce competing emotional cues rather than resolve absence-based sentiment. The substantial macro-F1 improvement from initialization (+0.435) highlights effective transfer from GoEmotions pre-training, while post-convergence instability indicates sensitivity to noise in multi-turn dialogue context.

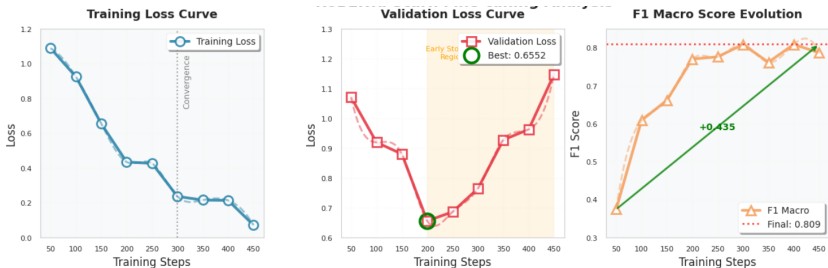

Figure 3: Training dynamics of RoBERTa-3turn for dialogue emotion classification. Training loss decreases rapidly and stabilizes by step 300. Validation loss reaches its minimum at step 300 (marked), after which overfitting emerges, motivating early stopping. Macro-F1 improves steadily (+0.435 overall), with strongest gains for positive emotion. Per-class trajectories indicate stable learning for positive and negative classes, while neutral performance remains comparatively volatile, reflecting contextual ambiguity introduced by multi-turn inputs.

## C    RESULTS DETAILS

### C.1    CONFUSION MATRIX ANALYSIS

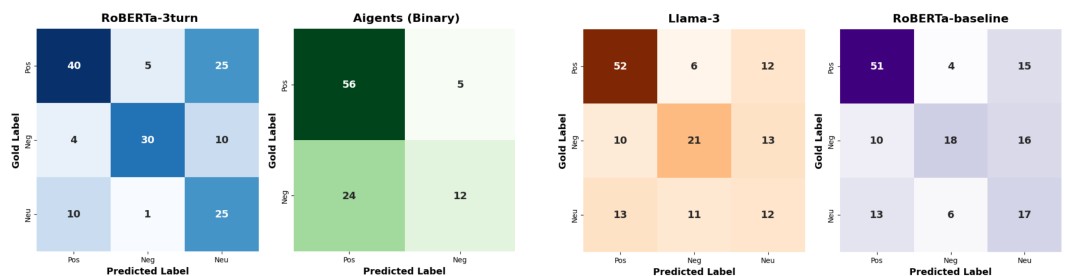

Figure 4: Comparison of model performance and human annotation consistency. (Left) Confusion matrix for the RoBERTa-3turn model. (Right) Inter-annotator agreement matrix ($\kappa = 0.579$) illustrating the distribution of concordant and discordant labels across the 150-sample gold standard.

Figure 4 summarizes confusion matrices for all evaluated models. Across paradigms, the dominant error pattern involves confusion between *neutral* and *positive* classes. In particular, neutral

utterances are frequently predicted as positive (e.g., 13/36 neutral instances for the RoBERTa baseline), reflecting the difficulty of distinguishing polite acknowledgment or factual engagement from genuine positive affect (e.g., "That's interesting"). Confusion between *negative* and *neutral* is less frequent (6–10 cases per model), consistent with negative emotions being more lexically explicit. Direct *positive–negative* confusion is rare (4–10 cases), indicating that emotional polarity is generally well captured when affect is overt; most errors instead involve neutral as an intermediate category.

On binary evaluation, Aigents exhibits asymmetric error behavior, with more false positives (negative→positive) than false negatives. This reflects lexicon sensitivity rather than contextual reasoning, leading to over-triggering in ambiguous cases and missed detections when sentiment is implicit.

## C.2 PER-CLASS ANALYSIS

Figure 2 summarizes per-class F1 scores across models, revealing consistent trends. **Positive emotion** is the easiest class for all approaches (F1: 0.630–0.750), reflecting the prevalence of explicit positive lexical cues in dialogue (e.g., expressions of appreciation or affiliation). Aigents achieves the highest positive F1 (0.750) on the binary subset, indicating that rule-based lexicons effectively capture common positive patterns. **Negative emotion** shows moderate performance (F1: 0.500–0.714), with RoBERTa-3turn outperforming other models. This suggests that multi-turn context aids the disambiguation of criticism and dissatisfaction from neutral statements, which are often lexically similar but pragmatically distinct. **Neutral emotion** remains the most challenging class for all three-class models (F1: 0.405–0.562), with Llama-3 performing worst (0.414). This difficulty reflects the absence-based nature of neutrality, which lacks distinctive surface markers and is subject to annotation ambiguity. Precision–recall patterns further differentiate models. Aigents exhibits high positive recall but lower precision, consistent with aggressive lexicon matching. RoBERTa-3turn maintains relatively balanced precision and recall across classes, while Llama-3 shows low neutral recall, indicating a tendency to default to polarized labels under uncertainty.

## C.3 ERROR ANALYSIS

We manually inspected 30 misclassified instances (10 per model) to identify recurrent failure modes.

**Implicit Emotion (42%):** All models struggle when affect is conveyed pragmatically rather than lexically. For example, "Why isn't Janine coming?" is labeled neutral despite an implied accusatory tone, and "Wow, really?" is ambiguous between enthusiasm and skepticism.

**Sarcasm and Irony (23%):** Utterances where surface polarity contradicts intended meaning (e.g., "Oh that's just great.") consistently cause errors across models, highlighting the limitations of both pattern-based and contextual encoders.

**Neutral–Positive Boundary (35%):** Polite or transactional expressions (e.g., "Thanks for the information") are frequently misclassified as positive, underscoring annotation ambiguity and the absence-based nature of neutrality.

**Model-Specific Trends:** Aigents fails when cross-turn or third-party context is required for correct attribution. RoBERTa occasionally overgeneralizes from lexical associations learned during pre-training (e.g., "Oh God"), while Llama-3 tends to assign polarized labels to ambiguous cases, likely reflecting bias toward emotionally salient content in its pre-training data.

## D AIGENTS GRAPH CONSTRUCTION

Figure 5 visualizes the dialogue-level social graph constructed from Aigents for a 10-turn conversation. Nodes represent dialogue participants, while directed edges encode inferred emotional attribution between speakers across turns. Edge color indicates polarity (green for positive, red for negative), and edge labels denote the corresponding confidence or strength score produced by the rule-based pipeline. This graph formulation makes explicit how emotional interactions accumulate across dialogue turns, allowing inspection of asymmetric or conflicting attributions between speaker pairs. Unlike neural encoders such as RoBERTa, which implicitly model interaction structure within

latent representations, Aigents exposes the interaction graph directly, enabling transparent analysis of emotion flow and speaker-to-speaker dynamics. In contrast, our RoBERTa models yield a structurally similar interaction pattern but additionally assign a neutral label, which is omitted here due to Aigents' binary formulation.

Figure 5: Dialogue-level social graph induced by Aigents for a 10-turn conversation. Nodes denote speakers and edges represent directed emotion attributions, and color indicating polarity (positive/negative) and edge labels showing attribution scores and polarity.

## E   INTER-ANNOTATOR AGREEMENT

To validate our evaluation benchmark, two independent annotators (with English proficiency and background in affective computing) labeled a 150-sample subset of the DialogRE dataset. We achieved a Cohen's Kappa of $\kappa = 0.579$ (see 6), representing "moderate" agreement according to standard interpretability scales Landis & Koch (1977). The disagreements, primarily centered on the distinction between neutral and subtle negative affect, indicating the inherent subjectivity of social-emotional relation extraction. All discordant instances were subsequently resolved through a third-party adjudication process to establish a final consensus gold standard for model benchmarking.

## F   ABLATION RESULTS

The Llama-3 ablation study (Table 3) reveals fundamental constraints on few-shot performance. The performance range across 9 configurations (F1 = 0.294 to 0.501, span = 0.207) demonstrates significant sensitivity to prompting strategy, yet even optimal configuration achieves only F1 = 0.501—substantially below both RoBERTa-3turn (F1 = 0.636) and Aigents Mode 3 (F1 = 0.624). This performance gap, combined with the degradation observed in alternative prompt templates (Chain-of-Thought mean = 0.328), suggests that few-shot prompting fundamentally underperforms task-specific fine-tuning or interpretable rule-based approaches on structured emotion classification tasks.

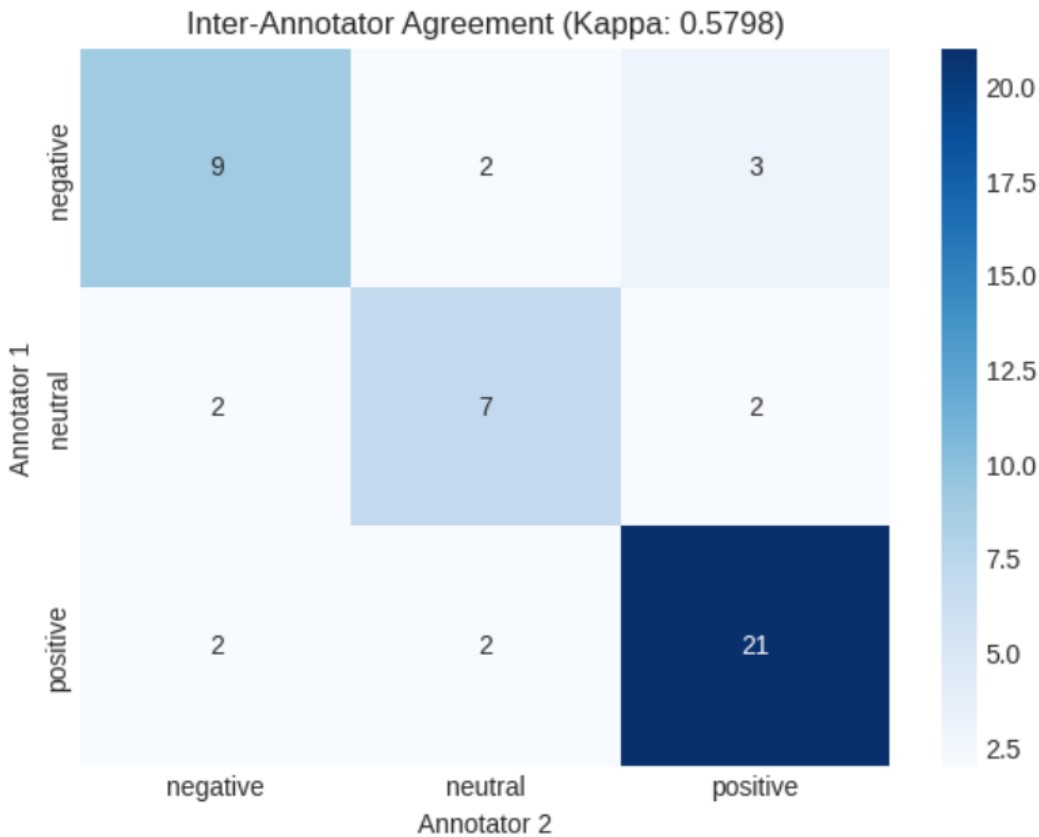

Figure 6: Inter-annotator agreement results with Cohen's Kappa ($\kappa$) interpretation. Our achieved $\kappa = 0.57$ falls in the "moderate" agreement range (0.41–0.60), validating annotation reliability.

Table 3: Llama-3-8B prompt engineering ablation study: F1 macro scores across 3 prompt templates and 3 shot counts (9 configurations total). Baseline configuration (Template 1, 10-shot) achieves F1 = 0.488. Best configuration (Template 1, 3-shot) achieves F1 = 0.501. Results demonstrate limited improvement potential through prompt engineering alone. All configurations evaluated on identical 150-sample test set.

| Template | 3-shot | 5-shot | 10-shot | Mean |
|---|---|---|---|---|
| 1. Standard | 0.501 | 0.447 | 0.488 | 0.479 |
| 2. Chain-of-Thought | 0.394 | 0.296 | 0.294 | 0.328 |
| 3. JSON-Format | 0.382 | 0.356 | 0.390 | 0.376 |
| **Mean (all)** | 0.426 | 0.366 | 0.391 | |

