# OpenReview forum: "Interpretable Emotion Attribution in Social Graphs: A Comparative Analysis of Rule-Based, Transformer, and LLM Models"
_mathai.club/MathAI/2026/Conference — 2026 Oral_

### Official Review · Reviewer_ykyV · 2026-03-11
**Technical Corrections: Interpretable Emotion Attribution in Social Graphs: A Comparative Analysis of Rule-Based, Transformer, and LLM Models**

**Rating:** 6
**Confidence:** 2

**Review:**

# Technical Corrections

1.	Formatting of references to sections: The text contains unfilled cross-references (e.g., "see Table ??" in lines 66 and 71). These need to be corrected to the appropriate table and figure numbers.

2.	In-text citation style: To improve readability, during the first and semantic citation of works, it is recommended to explicitly highlight the authors in the text, for example: (as shown by Dekker et al. (2019), Khemani et al. (2024)) (affects lines 35-36 and similar places throughout the text).

3.	Quality of illustrations: The image in Figure 5 (lines 760-780) has low resolution and poor quality. The authors should replace it with a higher resolution version suitable for printing.

4.	Standardization and formatting of the bibliography: The reference list needs to be brought into a uniform format. The following shortcomings were found in the current version:

a.	Several arXiv preprints are formatted as university technical reports (e.g., items 625-627, 629-631). The formatting for preprints should be standardized (e.g., using "arXiv:..."). It is recommended to check all such entries, specifically where an arXiv reference is given.

b.	Some entries (e.g., lines 617-623) list an excessively long number of authors. It is recommended to shorten these, using "et al." after the first 3-5 surnames to match the general style.

c.	It is desirable to add or verify the presence of DOI links for all articles where possible, and format them as active hyperlinks for the convenience of readers.

These comments do not concern the scientific value of the work and are aimed at improving its presentation and formatting.

P.S. This review is purely technical in its details. An additional expert assessment is required.

---

### Official Review · Reviewer_PEoH · 2026-03-11
**Review of "Interpretable Emotion Attribution in Social Graphs: A Comparative Analysis of Rule-Based, Transformer, and LLM Models"**

**Rating:** 6
**Confidence:** 4

**Review:**

This paper elaborates on a controlled comparison of three paradigms for emotion attribution in dialogue-based social graphs: an interpretable rule-based system (Aigents), a fine-tuned transformer (RoBERTa-large), and a few-shot LLM (Llama-3-8B). The task is formulated as predicting the emotional relation from one entity to another as positive, negative, or neutral, using local conversational context. On the reported benchmark, RoBERTa-3turn achieves the best overall Macro-F1 of 0.636, while the rule-based Aigents system remains competitive in restricted settings, and few-shot Llama-3 performs substantially worse.

The paper is interesting and well motivated. Its main strength is the comparison across fundamentally different modeling paradigms rather than across minor variations of the same architecture. The emphasis on interpretability is also valuable, especially for applications in social science where auditability matters. The paper has practical value and presents empirical observations.

At the same time, I have several concerns.

First, the paper does not appear to be a strong fit for a mathematics-oriented conference. The contribution is primarily empirical and engineering-oriented: dataset construction, prompting, fine-tuning, and comparison of model families. I do not see substantial mathematical innovation or theoretical development beyond a standard relation-classification setup. For a venue centered on mathematics or mathematical innovation, this weakens the paper’s positioning.

Second, the experimental reporting is not always clear enough. In particular, the Llama-3 results are confusing. The introduction reports a preliminary 10-shot F1 of 0.526, subsection "6.1 Overall performance" Macro-F1 = 0.528, while the systematic ablation across nine prompt configurations reports a best score of only 0.501. The manuscript should explicitly explain whether these numbers come from different setups, subsets, or stages of experimentation. So, this is the lack of clarity in experimental reporting.

Third, there is an internal inconsistency in the confidence intervals for the best model. Table 2 reports RoBERTa-3turn with Macro-F1 0.636 and CI [0.554, 0.715], whereas Section 6.2 reports the same model with CI [0.597, 0.673] (line 421). This inconsistency should be corrected.

Fourth, minor issue: there are no links to the tables in the “Introduction” (066 and 071 lines).

Overall, I think this is a good and useful paper with a clear applied contribution. The comparison between interpretable and black-box approaches is worthwhile, and the reported results are interesting enough to merit attention. However, the paper needs clearer and more consistent reporting of experimental conditions and technical mistakes (table links), and better alignment with the target venue (mathematics-focused conference).

---

### Official Review · Reviewer_jwvY · 2026-03-13
**Interesting research question, but underpowered test set (150 instances), questionable silver labeling, unfair LLM comparison, sitcom-only data, and broken table references.**

**Rating:** 5
**Confidence:** 3

**Review:**

This is an interesting research question: whether interpretable rule-based systems can compete with neural models for emotion attribution in social graphs. The comparison across three paradigms (Aigents, RoBERTa, Llama-3-8B) is well-motivated for high-stakes domains requiring transparency. The finding that few-shot Llama-3-8B significantly underperforms fine-tuned models is a valuable counter-intuitive contribution worth publishing.

However, the execution has significant issues. The test set of 150 instances after filtering is too small for reliable statistical claims about "indistinguishability" between systems. The silver labeling approach is paradoxical: training data is labeled by the same LLM that achieves only F1=0.526, raising questions about label quality. The comparison between 10-shot prompting and fine-tuning on ~1,200 examples is not compute-fair. DialogRE consists of scripted sitcom dialogue (Friends), which differs substantially from real social media or forensic conversations the authors cite as motivating applications; however, this was addressed in limitations. The terminology is also problematic - classifying positive/negative/neutral is sentiment analysis, not emotion attribution. Finally, broken table references ("see Table ??") indicate incomplete preparation.

Areas for improvement:
• Expand evaluation set for adequate statistical power
• Add a second dataset with real (non-scripted) social data
• Provide fair LLM comparison (fine-tuned or substantially more shots)
• Describe Aigents rules in sufficient detail for reproducibility
• Fix presentation errors
• Clarify sentiment vs. emotion terminology

---

### Decision · Program_Chairs · 2026-03-14

**Decision:**

Accept (Oral)

**Comment:**

Dear Author(s),

On behalf of the Program Committee of the International Conference on Mathematics of Artificial Intelligence (MathAI 2026), we are pleased to inform you that your paper has been accepted for an oral presentation at MathAI 2026.

Your paper was evaluated through a rigorous two-stage review process involving both automated screening and expert review by members of the Program Committee. The reviewers recognized the quality and contribution of your work.

Presentation details:

- Format: Oral presentation (15–20 minutes + 5 minutes Q&A)
- Mode: You may present either in person (offline) at the conference venue in Sirius, Russia, or remotely via Zoom. Please indicate your preferred mode when confirming your participation.
- Conference dates: Marh 30 - April 3, 2026
- Website: https://mathai.club

Next steps:

1. Please confirm your participation and presentation mode by replying to this email mathai.club@yandex.ru no later than March 15, 2026 18:00 Moscow time.
2. If you plan to attend in person, the organizing committee will provide accommodation details separately.
3. Please prepare your final camera-ready manuscript according to the formatting guidelines available at https://mathai.club and upload it to OpenReview by March 15, 2026 18:00 Moscow time.

Should you have any questions regarding the program, logistics, or your presentation slot, please do not hesitate to contact us.

We look forward to your contribution to MathAI 2026.

With kind regards,

MathAI 2026 Program Committee
International Conference on Mathematics of Artificial Intelligence
https://mathai.club
OpenReview: https://openreview.net/group?id=mathai.club/MathAI/2026/Conference
Telegram: https://t.me/MathAI_club
Email: mathai.club@yandex.ru